# CoT-RC: Chain-of-Thought Reflection and Correction for Image Generation without Extra Training

## Abstract

Recent studies have explored integrating Chain-of-Thought (CoT) reasoning into image generation to improve accuracy and controllability. However, existing methods either rely on costly training, separate reasoning from generation, or lack fine-grained visual error correction. We propose a training-free CoT-enhanced image generation framework that leverages the semantic understanding and positional awareness of Unified Multimodal Models (UMMs). Our method introduces a CoT-guided Reflection Module for image-level global correction and a semantic-driven token-level local correction module for fine-grained refinement, forming a dynamic reasoning loop with iterative triggers and backtracking. Experiments demonstrate that our approach improves the Show-o baseline from 68% to 78% on GenEval and achieves a 14% gain on T2I-CompBench, outperforming prior CoT-based methods under the same baseline, including reinforcement learning-based approaches. Our framework is entirely training-free, efficient, and establishes a new paradigm for CoT in image generation.

## 1 Introduction

With the increasing scale of model and training data, Large Language Models(LLMs), e.g., GPT-3(Brown et al. (2020)), PaLM(Chowdhery et al. (2022)), have achieved remarkable performance on mutiple general tasks. However, their capabilities in complex tasks such as mathematical computation and programming remain limited. To address this limitation, the Chain-of-Thought (CoT Wei et al. (2022)) paradigm is proposed, enabling LLMs to generate intermediate reasoning steps and thereby significantly improving performance (e.g., GPT-5 Wang et al. (2025), DeepSeek-R1 Guo et al. (2025a)). Inspired by this success, recent work has explored integrating CoT into image generation to enhance both accuracy and controllability. For instance, PARM (Guo et al. (2025b)) introduces stepwise rewards via reinforcement learning to dynamically guide sampling in autoregressive generation, improving semantic alignment and fine-grained details. Nevertheless, it is challenged by substantial computational costs during training, dependency on subjectively defined reward structures. ImageGen-CoT(Liao et al. (2025)) employs supervised fine-tuning (SFT) to integrate textual CoT, allowing Mutimodal Large Language Modeels (MLLMs) to generate reasoning details before image synthesis. However, it separates reasoning from generation and depends on large-scale, high-quality CoT data. PromptCoT (Yao et al. (2024)) extends the prompts with textual CoT in a low-cost, zero-training manner, but lacks reasoning about visual tokens and fails to correct local errors, limiting improvements in fine-grained visual details.

End-to-end Unified Multimodal Models (UMMs) have recently been proposed to unify the content understanding and visual generation, which are able to analyze semantic information from multimodal inputs, implicitly capture spatial relationship of objects and cross-modal correspondences during inference. These inherent capabilities suggest a natural potential to integrate CoT-based semantic feedback and local correction directly into the image generation process without training. However, existing methods underutilize this potential, relying primarily on semantic understanding for image regeneration and failing to exploit the intrinsic positional awareness inherent in UMMs.

Motivated by this, we propose a **training-free CoT-enhanced image generation framework**. Our framework leverages the semantic understanding and positional awareness of UMMs by integrat-

ing CoT into end-to-end UMMs, enabling dynamic **image-level global correction** through sematic understanding and reasoning, while simultaneously supporting **token-level local correction** based on positional awareness. Specifically, we develop a CoT guided reflection module (CoT-Reflection module) to perform reasoning over generated intermediate visual tokens, automatically identify semantic deviations, and generate refined textual prompts for image-level global semantic correction without requiring additional training. A Semantic-driven token-level local correction module is designed to leverage the positional awareness ability to generate masks for erroneous regions, preserve correct visual tokens, and mask the identified erroneous tokens for resampling guided by refined textual prompts, allowing fine-grained token-level local correction with minimized computation cost. Moreover, we design a CoT-guided dynamic reflection and correction mechanism that integrates the CoT-Reflection module for global semantic correction and the token-level local correction module for fine-grained error refinement. The designed trigger points and backtracking in this reflection and correction mechanism enable these modules to operate iteratively, ensuring global semantic consistency while efficiently correcting local errors.

Extensive experimental results show that our training-free approach improves the baseline model (Show-o(Xie (2025)) is used in our experiments) from 68% to 78% on the GenEval(Ghosh et al. (2023)) benchmark, achieves stronger semantic alignment and finer generation control in complex scenarios. In addition, our method improves the average score on the T2I-CompBench benchmark by **14%** compared to the baseline model. Unlike prior CoT-based methods that depend on reinforcement learning or external discriminators, our method is entirely training-free and cost-effective. Moreover, it surpasses reinforcement learning-based CoT methods (e.g., PARM) built on the same baseline on both benchmarks, establishing a new paradigm of CoT in image generation.

The main contributions of this work are as follows:

- We propose a training-free CoT-enhanced image generation framework, unifying reasoning and synthesis within end-to-end UMMs.
- A CoT-guided dynamic Reflection and Correction mechanism is proposed, which includes CoT-guided global image-level correction and semantic-driven token-level local correction, forming a dynamic reasoning loop to ensure both semantic consistency and fine-grained generation control.
- Extensive experiments demonstrate the substantial improvements of the proposed method compared to the baseline and state of the arts, validating the effectiveness and efficiency of our method.

## 2 RELATED WORK

**Chain-of-Thought in Language Models.** The Chain-of-Thought (CoT) paradigm, introduced by Wei et al. (2022), generates intermediate reasoning steps in large language models (LLMs), significantly improving performance on tasks such as mathematical computation and programming. Subsequent studies have extended CoT in diverse directions. Self-Consistency(Wang et al. (2023)) enhances reasoning robustness through diverse sampling, Tree-of-Thought(Yao et al. (2023a)) models reasoning as a tree-based search to improve decision-making, and ReAct(Yao et al. (2023b))combines reasoning with external tool usage, broadening CoT's application scope. These studies show that explicit stepwise reasoning improves model transparency, interpretability, and task accuracy, establishing CoT as a key paradigm for enhancing LLM reasoning.

**Unified Multimodal Models for Understanding and Generation.** With advances in LLMs, multimodal models (UMMs) have emerged, extending language models to jointly process images and text. For understanding tasks, representative works include BLIP-2(Li et al. (2023)) and LLaVA(Liu et al. (2023)), which couple visual encoders with language models to achieve strong vision–language alignment. For generation, UMMs aim to enable bidirectional image–text synthesis. Existing approaches can be categorized into three major types. The first is **diffusion-based methods**, such as Dual Diffusion(Li et al. (2025)) and UniDisc(Bao et al. (2025)), which integrate multimodal constraints into the diffusion process. The second is **autoregressive methods**, such as Chameleon(Team (2024a)), Emu3(Wang et al. (2024)), and Liquid(Chen et al. (2024)), which predict discrete image tokens end-to-end. The third is **hybrid AutoregRessive(AR) + diffusion methods**, such as Transfusion(Zhou et al. (2024)) and Show-o(Xie (2025)), which combine the benefits of both approaches.

Some approaches rely on external generators (e.g., Emu(Sun et al. (2023)) predicts CLIP(Radford et al. (2021)) features and renders them via Stable Diffusion(Rombach et al. (2022))), while others generate discrete tokens directly within the UMM.Overall, these methods highlight the potential of UMMs as a unified framework for bridging visual understanding and generation.

**Chain-of-Thought in Vision.** In the visual domain, CoT has been applied to both understanding and generation tasks. For understanding, methods such as Multimodal-CoT(Zhang et al. (2023)) and Visual-CoT(Shao et al. (2024)) extend textual CoT reasoning to visual question answering and image reasoning, improving interpretability and accuracy. Benchmarks include VQA(Goyal et al. (2017)) and OK-VQA(Marino et al. (2019)). In visual generation, CoT improves accuracy, especially in autoregressive settings. PromptCoT(Yao et al. (2024)) extends prompts with textual CoT before generation for low-cost, zero-training enhancement, while ImageGen-CoT(Liao et al. (2025)) injects textual reasoning chains via supervised learning. Both operate primarily at the text level, lacking interaction with visual tokens, limiting performance in complex scenarios. PARM(Guo et al. (2025b)) guides autoregressive generation with stepwise rewards, and T2I-R1(Jiang et al. (2025)) combines semantic- and token-level CoT for high-level planning and block-wise generation. However, many recent CoT-based approaches for visual generation either operate mainly at the textual prompt level (e.g., PromptCoT Yao et al. (2024)), or rely on substantial additional training, fine-tuning, or reward-model design (e.g., ImageGen-CoT Liao et al. (2025), PARM Guo et al. (2025b), T2I-R1 Jiang et al. (2025)). While some methods have begun to combine semantic- and token-level CoT, they typically incur significant training or RL costs. Consequently, the potential of UMMs' intrinsic semantic reasoning and positional awareness for efficient, training-free token-level correction remains underexplored.

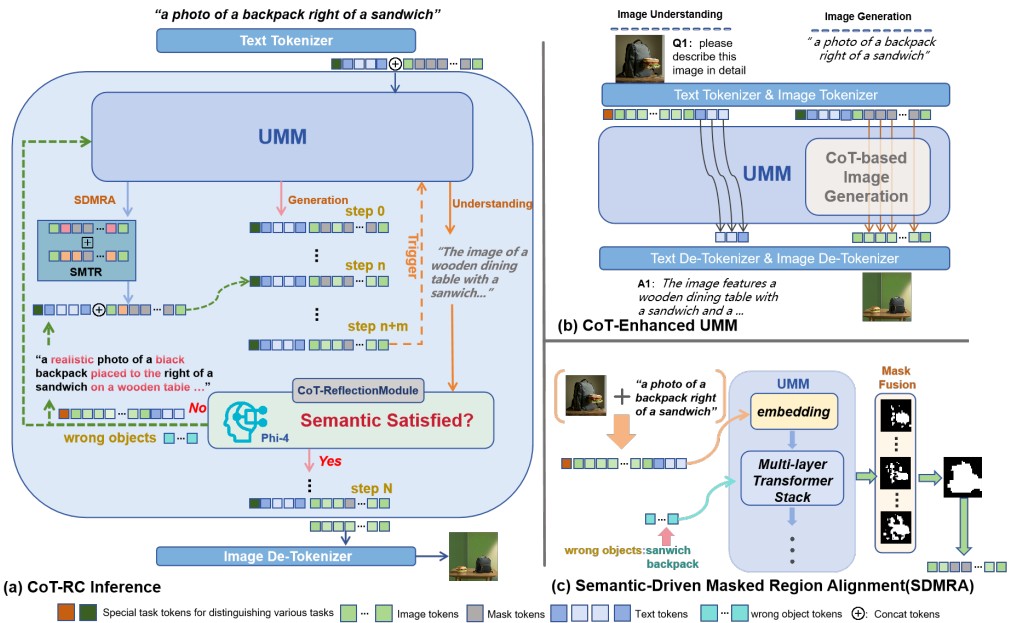

Figure 1: Chain-of-Thought (CoT) Enhanced Image Generation Framework. SMTR (Semantic Masked Token Retention) covers erroneous regions during correction while retaining correct tokens to ensure generation stability. At Step n+m, descriptions generated by the UMM from intermediate visual tokens are combined with the original prompt and fed into the CoT-Reflection module, which analyzes deviations, identifies erroneous objects, assigns relevance scores, and produces a refined prompt if needed. SDMRA localizes erroneous token regions and generates masks while retaining partially correct tokens. The masked intermediate visual tokens are then backtracked to Step n and resampled under the guidance of the refined prompt, correcting errors while preserving global consistency.

# 3 METHODOLOGY

## 3.1 MODEL FRAMEWORK

Figure 1(a) presents the propose training-free CoT-enhanced image generation framework, which achieves efficient and controllable image generation in complex scenarios by integrating an **image-level global correction** based on sematic understanding and reasoning and **token-level local correction** based on UMMs' positional awareness. The framework consists of two core modules. The Chain-of-Thought Guided Reflection module (CoT-Reflection module) takes the original text prompt and descriptions of intermediate visual tokens generated by UMMs as input, performs step-by-step reasoning to analyze semantic deviations, generates refined prompts, and detects wrongly generated objects for token-level correction. The Semantic-Driven Masked Region Alignment (SDMRA) module localizes the regions of the wrongly generated objects with the guidance of the cross-attention map. Instead of treating correction as a heuristic post-processing, this module detects semantic errors and generates precise region masks for local error correction, enabling fine-grained and semantically consistent correction in the generation process.

We design a CoT-guided dynamic reflection and correction mechanism that combines global semantic correction with the CoT-Reflection module and fine-grained local correction with the SDMRA module. In this mechanism, trigger points and backtracking are designed, which allow these two modules to operate iteratively and form a loop that ensures global semantic consistency while accurately and efficiently correcting local errors.

The iterative pipeline proceeds with the following steps, Intermediate visual token generation → Trigger → Understanding → CoT-Reflection → SDMRA → SMTR with refined prompt → Backtrack & Resample, until the generated image satisfies semantic constraints or reaches the iteration limit.

### 3.1.1 CHAIN-OF-THOUGHT GUIDED REFLECTION(CoT-REFLECTION) MODULE

To enable semantic evaluation during image generation, we design the CoT-Reflection Module, which performs structured reasoning over generated intermediate visual tokens. As shown in Figure 1(a), the module takes the original prompt and the UMM's description of the intermediate generation as input. To ensure generalization and consistency, it employs structured CoT reasoning via Phi-4, integrating LLM reasoning with semantic constraints to produce comprehensive outputs, including error analyses, identified deviant objects, and refined prompts that guide subsequent token-level corrections.

As illustrated in Figure 2, the CoT-Reflection Module performs structured reasoning over intermediate visual tokens in a **five-step** process guided by predefined semantic evaluation rules. In **Step 1**, the module compares the generated visual token descriptions with the original prompt, focusing on explicitly specified hard constraints such as required objects, exact quantities, constrained attributes (color, size, position), and scene context. Deviations are categorized into missing objects, incorrect quantities, incorrect attributes, and scene mismatches, with deviant objects precisely identified.

In **Step 2 (Error Object Analysis)**, the module collects the identified deviant objects and associated errors from Step 1, providing the basis for subsequent token-level error region localization and local correction. In **Step 3 (Relevance Score)**, a relevance score from 0 to 1 is assigned according to a set of designed rules that consider the presence, quantity, attributes, and scene consistency of objects, reflecting the alignment between the generated tokens and the prompt. A score of 1 indicates full compliance with all hard constraints, while lower scores correspond to increasing numbers or severity of violations.

In **Step 4 (Refinement Decision)**, the module determines whether prompt refinement is necessary based on the relevance score and the recorded errors. Finally, in **Step 5 (refine Prompt)**, a refined prompt is generated to correct the identified violations, including counts, attributes, and relative positions, while preserving all compliant information and integrating additional non-conflicting details from the generated description. The output is designed to guide downstream token-level local corrections and is concise, interpretable, and consistent with the original prompt.

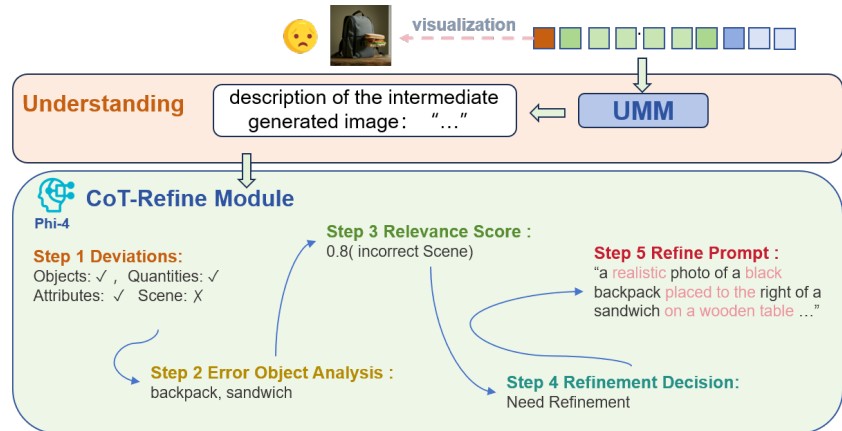

Figure 2: CoT-Reflection Module performs structured reasoning over intermediate visual tokens through five steps, including (1) identifying deviations from hard constraints in the prompt, (2) collecting deviant objects, (3) assigning a relevance score quantifying alignment with the prompt, (4) deciding whether prompt refinement is needed, and (5) generating a refined prompt to guide token-level corrections.

### 3.1.2 SEMANTIC-DRIVEN MASKED REGION ALIGNMENT(SDMRA) MODULE

End-to-end UMMs inherently encode rich semantic and spatial information, capturing object relationships, spatial layouts, and cross-modal correspondences during inference. Existing methods, however, primarily exploit UMMs for global semantic understanding, underutilizing their fine-grained positional awareness. To fully leverage these capabilities, our **SDMRA module** integrates UMM attention-based localization with CoT-guided analysis, enabling precise token-level corrections without requiring additional discriminators or reward models.

As shown in Figure 1(c), when a trigger point detects semantic deviations via the CoT-Reflection Module, SDMRA leverages the error objects identified along with the UMM's visual perception to localize erroneous token regions and performs targeted corrections guided by the refined prompt. To improve localization accuracy, SDMRA fuses attention maps from multiple Transformer layers. Lower layers provide fine-grained local details, while higher layers capture global semantic relations Raghu et al. (2021); Abnar & Zuidema (2020),. The cross-layer integration ensures robust and precise identification of error regions.

Let $\mathcal{L} = \{10, 11, 14, 16, 17, 18\}$ denote the selected Transformer layers of the Show-o model. For each layer $l \in \mathcal{L}$ and token position $i$, attention scores $A_i^{(l)}$ are normalized and positions exceeding a threshold $\tau_l$ are collected as candidate error tokens. The union of these attention-based candidates across layers is denoted as $M_{\mathrm{attn}}$.

Error objects identified by CoT-Reflection Module, $\mathcal{O}_{\mathrm{err}}$, are mapped to token indices $M_{\mathrm{err\_obj}}$. The final error mask is obtained by combining attention-based and CoT-Reflection signals:

$$\mathcal{M} = M_{\mathrm{attn}} \cup M_{\mathrm{err\_obj}}. \tag{1}$$

Tokens in $\mathcal{M}$ are masked for resampling, while partially correct tokens are retained. The refined prompt is concatenated at the backtrack state to guide correction. This process iterates until semantic compliance is achieved or a maximum iteration limit is reached. By leveraging the UMM's attention and semantic understanding, SDMRA enables efficient, fine-grained, token-level correction without any additional discriminators or reward models.

### 3.2 COT-GUIDED DYNAMIC REFLECTION AND CORRECTION(COT-RC) MECHANISM

As illustrated in Figure 1(a), the proposed training-free CoT-enhanced framework incorporates a dynamic reflection and correction mechanism to iteratively refine image generation. To validate its effectiveness, we apply this mechanism on the Show-o model(Xie (2025)), which generates intermediate images using a MaskGIT-style sampling process. Trigger points are inserted in the mid-to-late

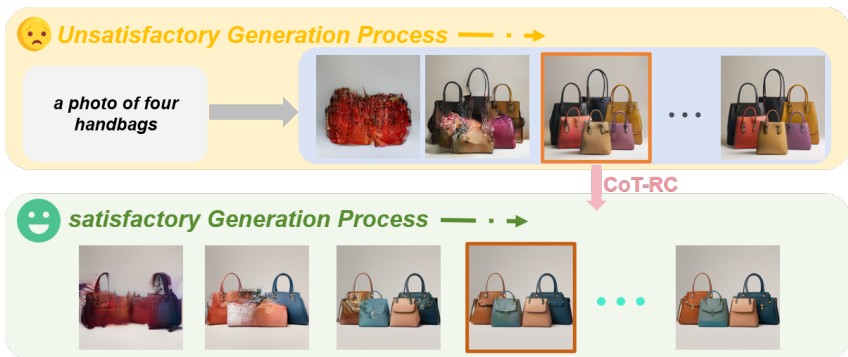

Figure 3: Effect of CoT-RC integration on image generation comparing before and after

stages of generation to evaluate intermediate representations. The CoT-Reflection Module performs multi-step semantic reasoning to identify inconsistencies between generated tokens and the prompt (e.g., object categories, counts, or spatial relations), producing refined prompts to correct semantic deviations. The process then backtracks to key steps, applies SDMRA masks to erroneous token regions while retaining partially correct tokens, and resamples under the guidance of the refined prompts, thus avoiding full-image regeneration. This iterative loop—"Generation → Evaluate → Reflection → Backtrack → Resample"—continues until semantic constraints are satisfied or the iteration limit is reached, enabling efficient error correction while preserving global semantic consistency. Although we demonstrate this mechanism on Show-o, it is model-agnostic and can be integrated into other autoregressive generation frameworks.

The advantages of the mechanism are threefold. First, leveraging the UMM's semantic understanding and attention, the token-level correction loop is triggered at intermediate checkpoints and backtracks to the designated step for targeted resampling, allowing focused refinement while preserving global semantic consistency. Second, the method is entirely training-free, requiring no additional discriminators or reinforcement learning, enhancing reproducibility and reducing reliance on large-scale data and computation. Third, it is robust in complex scenarios, preserving global semantic control while enabling local detail correction, especially for images containing multiple objects, attributes, and relationships. As shown in Figure 3, the entire generation process enabled by the CoT-RC mechanism significantly improves the accuracy of image generation.

## 4 EXPERIMENTS

### 4.1 EXPERIMENTAL SETUP

To evaluate the effectiveness of the proposed **training-free CoT-RC mechanism** for image generation, we conducted experiments on two benchmarks, **GenEval**(Ghosh et al. (2023)) and **T2I-CompBench**(Ding et al. (2023)). GenEval evaluates the compositional capabilities of text-to-image models by breaking prompts into object-level tasks and performing fine-grained, instance-level assessment using object detectors. T2I-CompBench evaluates open-world compositional tasks involving multiple objects, attributes, and spatial relationships, covering three categories: object attributes (Color/Shape/Texture), object relationships (Spatial/Non-Spatial), and complex compositions (e.g., multi-object interactions). With prompts drawn from 12,000 real-world examples, it emphasizes generalization to unseen, non-predefined combinations.

All experiments were performed on a single NVIDIA A800 GPU. Generated image resolutions were set to $512 \times 512$ for Show-O. During SDMRA Module, we leveraged attention maps from layers 10, 11, 14, 16, 17, and 18 of the Show-O model, and set the maximum number of backtracking iterations to 10.

| Method | #Params | Single object | Two object | Counting | Colors | Position | Attribute binding | Overall |
|---|---|---|---|---|---|---|---|---|
| **Diffusion Models** | | | | | | | | |
| DALL-E2(Ramesh et al. (2022)) | 6.5B | 0.94 | 0.66 | 0.49 | 0.77 | 0.10 | 0.19 | 0.52 |
| DALL-E3(Betker et al. (2023)) | - | 0.96 | 0.87 | 0.47 | 0.83 | 0.43 | 0.45 | 0.67 |
| SDv2.1(Rombach et al. (2021)) | 0.9B | 0.98 | 0.51 | 0.44 | 0.85 | 0.07 | 0.17 | 0.50 |
| SDXL(Podell et al. (2023)) | 2.6B | 0.98 | 0.74 | 0.39 | 0.85 | 0.15 | 0.23 | 0.55 |
| SD3(Team (2024b)) | 2B | 0.99 | 0.94 | 0.72 | 0.89 | 0.33 | 0.60 | 0.74 |
| FLUX.1Black Forest Labs et al. (2025) | 12B | 0.99 | 0.81 | 0.79 | 0.74 | 0.20 | 0.47 | 0.67 |
| PixArt-$\alpha$(Chen et al. (2023)) | 0.6B | 0.98 | 0.50 | 0.44 | 0.80 | 0.08 | 0.07 | 0.48 |
| D-DiT(Li et al. (2025)) | 2B | 0.97 | 0.80 | 0.54 | 0.76 | 0.32 | 0.50 | 0.65 |
| **AutoRegressive Models** | | | | | | | | |
| Show-o(Xie (2025)) | 1.3B | 0.98 | 0.86 | 0.64 | 0.83 | 0.27 | 0.48 | 0.68 |
| Emu3(Wang et al. (2024)) | 8B | - | - | - | - | - | - | 0.66 |
| MUSE-VL(Xie et al. (2024)) | 7B | - | - | - | - | - | - | 0.57 |
| Show-o2(Xie et al. (2025)) | 1.5B | 0.99 | 0.86 | 0.55 | 0.86 | 0.46 | 0.63 | 0.73 |
| Show-o2(Xie et al. (2025)) | 7B | 1.00 | 0.87 | 0.58 | 0.92 | 0.52 | 0.62 | 0.76 |
| Janus-Pro(Chen et al. (2025)) | 7B | 0.99 | 0.89 | 0.59 | 0.90 | 0.79 | 0.66 | 0.80 |
| Show-O+PARM(Guo et al. (2025b)) | 1.3B | 0.99 | 0.86 | 0.64 | 0.83 | 0.66 | 0.64 | 0.77 |
| **Show-o+CoT-RC(Ours)** | 1.3B | 0.99 | 0.93 | 0.83 | 0.84 | 0.53 | 0.59 | 0.78 |

Table 1: Evaluation on the GenEval(Ghosh et al. (2023)) benchmark.The best score is in red, with the second-best in blue and third-best in green. #Params indicates the number of parameters of base LLM.“-”denotes that the results of the approaches are not reported in their paper.

## 4.2 COMPARISON EXPERIMENT

**Results on GenEval.** As shown in Table 1, on the GenEval benchmark, Show-o+CoT-RC achieves significant improvements across all six metrics, with an overall gain of approximately **10%**. Unlike global metrics such as CLIPScore or FID, GenEval emphasizes object relationships, attribute binding, and compositional reasoning rather than overall image quality. Under the same baseline, our training-free CoT approach outperforms reinforcement learning-based methods such as PARM. Notably, the Counting task improves by **+19%** and the Position task by **+26%**, highlighting that token-level local correction guided by global CoT-based reflection effectively enhances semantic understanding and spatial reasoning. Moreover, despite having only 1.3B parameters, Show-o with CoT-RC surpasses FLUX.1 (**nearly ten times larger in parameters**) and reaches performance comparable to autoregressive UMMs with roughly five times more parameters (e.g., Janus-Pro-7B), even exceeding them on certain tasks. These results demonstrate that integrating CoT-guided reflection with token-level correction substantially improves the model's compositional reasoning ability, enabling it to robustly handle complex multi-object scenes with accurate object-level alignment.

**Results on T2I-CompBench.** Similarly, on the T2I-CompBench benchmark (Table2), CoT-RC CoT-RC achieves significant improvements across all six metrics compared with the baseline, with an average gain exceeding **14%**. Notably, the Color, Texture, and Spatial tasks show the largest improvements, reaching **20%, 24%, and 19%**, respectively. Whether compared with larger mainstream diffusion models or autoregressive UMMs (e.g., Janus-Pro), our method demonstrates highly competitive performance across all metrics, surpassing Flux.1, Janus-Pro, and PARM on average. This indicates that CoT-RC effectively enhances semantic consistency and fine-grained attribute control in image generation.Furthermore, we observe that CoT-RC generally outperforms Janus-Pro on T2I-CompBench, while underperforming on GenEval. This discrepancy reflects the different emphases of the two benchmarks: GenEval focuses on fine-grained pairwise object relations, whereas T2I-CompBench emphasizes overall generation quality in complex multi-object scenes. By leveraging UMMs' global understanding for semantic refinement, CoT-RC prioritizes global optimization, maintaining coherent layout and semantic alignment in complex scene generation.

Overall, these findings suggest that CoT-RC, by combining image-level global semantic correction with token-level local correction, effectively incorporates multimodal understanding into image generation, thereby enhancing both generation quality and reasoning consistency without additional training.

| Method | Attribute Binding | | | Object Relationship | | Complex↑ |
|---|---|---|---|---|---|---|
| | Color ↑ | Shape↑ | Texture↑ | Spatial↑ | Non-Spatial↑ | |
| **Diffusion Models** | | | | | | |
| StructureDiffusion(Feng et al. (2023)) | 0.4990 | 0.4218 | 0.4900 | 0.1386 | 0.3111 | 0.3355 |
| Composable Diffusion(Liu et al. (2022)) | 0.4063 | 0.3299 | 0.3645 | 0.0800 | 0.2980 | 0.2988 |
| Attend-and-Excite(Chefer et al. (2023)) | 0.6400 | 0.4517 | 0.5963 | 0.1455 | 0.3109 | 0.3401 |
| PixArt-$\alpha$(Chen et al. (2023)) | 0.6690 | 0.4927 | 0.6477 | 0.2064 | 0.3197 | 0.3433 |
| CoMat(Jiang et al. (2024)) | 0.7827 | 0.5329 | 0.6468 | 0.2428 | 0.3187 | 0.3680 |
| SDXL(Podell et al. (2023)) | 0.5879 | 0.4687 | 0.5299 | 0.2131 | 0.3119 | 0.3247 |
| FLUX.1(Black Forest Labs et al. (2025)) | 0.7407 | 0.5718 | 0.6922 | 0.2863 | 0.3127 | 0.3703 |
| **AutoRegressive Models** | | | | | | |
| Show-o(Xie (2025)) | 0.56 | 0.41 | 0.46 | 0.20 | 0.30 | 0.29 |
| EMU3(Wang et al. (2024)) | 0.7544 | 0.5706 | 0.7164 | - | - | - |
| Janus-Pro-7B(Chen et al. (2025)) | 0.6359 | 0.3528 | 0.4936 | 0.2061 | 0.3085 | 0.3559 |
| Show-o + PARM(Guo et al. (2025b)) | 0.75 | 0.56 | 0.66 | 0.29 | 0.31 | 0.37 |
| **Show-o+CoT-RC(Ours)** | 0.7618 | 0.5375 | 0.7066 | 0.3959 | 0.3130 | 0.3549 |

Table 2: Evaluation on the T2I-CompBench(Ding et al. (2023)) benchmark.The best score is in red ,with the second-best in blue and third-best in green. "-"denotes that the results of the approaches are not reported in their paper. "↑" shows consistent performance improvements compared with the baseline.

| Model | Single object | Two object | Counting | Colors | Position | Attribute binding | Overall |
|---|---|---|---|---|---|---|---|
| Show-o(Xie (2025)) | 0.98 | 0.86 | 0.64 | 0.83 | 0.27 | 0.48 | 0.68 |
| **Show-o+CoT-Reflection(only)** | 0.99 | 0.89 | 0.70 | 0.85 | 0.34 | 0.57 | 0.72 |
| **Show-o+Simple CoT-Reflection+correction** | 0.98 | 0.88 | 0.71 | 0.82 | 0.30 | 0.50 | 0.70 |
| **Show-o+CoT-RC** | 0.99 | 0.93 | 0.83 | 0.84 | 0.53 | 0.59 | 0.78 |

Table 3: Evaluation on the GenEval(Ghosh et al. (2023)) benchmark. The best score is in red."CoT-Reflection (only)" uses only the CoT-Reflection Module without token-level correction. "Simple CoT-Reflection+correction" uses token-level correction with a simplified CoT-Reflection that skips prompt refinement and reduces reasoning steps. "CoT-RC" applies the full CoT-Reflection with structured reasoning and token-level local correction.

## 4.3 ABLATION STUDIES

The ablation results in Table 3 validate the contributions of each component in our CoT-RC framework. Incorporating only the **CoT-Reflection Module** already improves performance over the baseline, demonstrating the effectiveness of global reasoning for identifying and correcting semantic deviations. Adding token-level correction with a simplified CoT-Reflection further improves certain metrics, such as Counting, but yields limited gains in others, indicating that coarse reasoning alone is insufficient for robust correction. The full **CoT-RC** approach, combining structured multi-step CoT reasoning with token-level local correction, consistently achieves the best performance across tasks, particularly in complex reasoning benchmarks such as Counting and Position. This confirms that the synergy between global semantic understanding and fine-grained local corrections is critical for improving both overall generation quality and reasoning consistency without additional training.

## 4.4 VISUAL GENERATION

As shown in Figure 4, we conducted a qualitative comparison of visual generation under the same prompt in different models, including the Show-o baseline, the CoT-based PARM image generation method integrated with Show-o (Show-o + PARM), and the Janus-Pro-7B, the larger autoregressive UMM. As shown in Figure 4, Show-o can produce coherent scene layouts, but often suffers from inaccuracies in object counts, attributes, and positions in complex multi-object scenarios. Show-o+PARM shows improved performance in spatial relationships, but still struggles with generating the correct object quantities. Janus-Pro-7B excels in strict positional and relational accuracy, yet exhibits limitations in maintaining global semantic consistency and fine-grained attribute details.

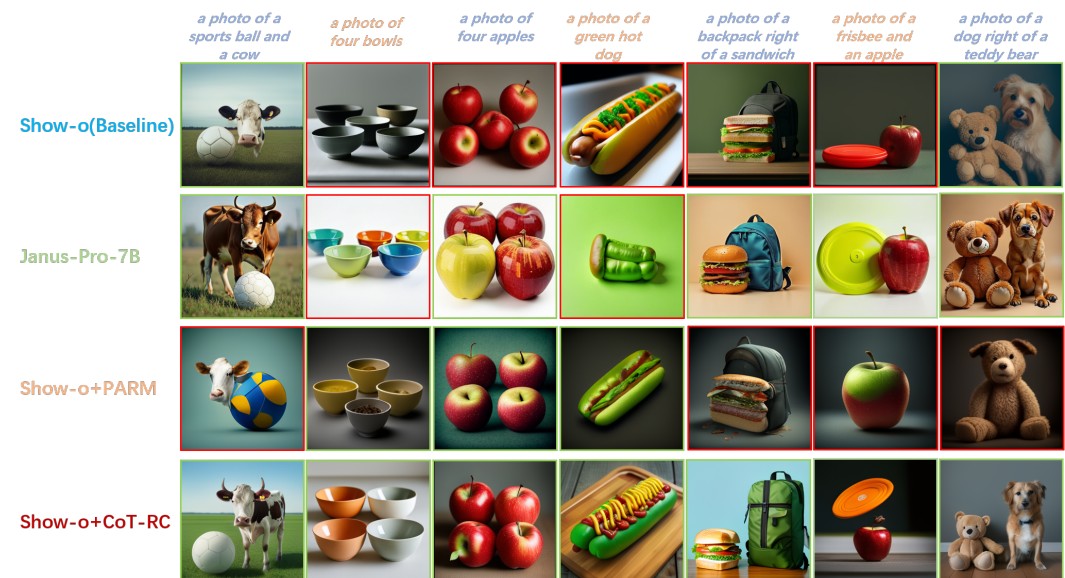

Figure 4: **Visualization results**. We show image generation results for the same prompt with four models, Show-o baseline Xie (2025), Show-o with PARM Guo et al. (2025b), Janus-Pro-7B Chen et al. (2025), and Show-o with **our CoT-RC mechanism**. Red boxes indicate incorrect generations and green boxes indicate correct ones.

In contrast, Show-o+CoT-RC delivers consistently superior results, particularly in complex multi-object scenes, where generated images better align with the given prompt and exhibit more precise details. This improvement comes from the synergy of the CoT-RC mechanism of global semantic understanding and fine-grained local correction. The CoT reflection module conducts step-by-step reasoning to detect semantic deviations and generate refined prompts, providing explicit guidance for correction. The Semantic-Driven Masked Region Alignment (SDMRA) Module leverages multi-layer Transformer attention to precisely locate error regions and apply targeted corrections. This global–local collaboration effectively addresses limitations of baseline and comparison methods, significantly enhancing both the accuracy and consistency of generated images.

## 5 CONCLUSION

We introduce a training-free Chain-of-Thought Reflection and Correction (CoT-RC) mechanism that enhances autoregressive image generation through token-level local corrections guided by semantic reasoning. CoT-RC leverages the intrinsic semantic understanding and positional awareness of unified multimodal models (UMMs) to detect generation deviations, refine prompts, and selectively resample erroneous regions without regenerating the entire image. Extensive benchmark evaluations demonstrate that CoT-RC significantly improves semantic alignment, spatial consistency, and overall generation quality compared with conventional CoT methods applied to the same base model and other strong baselines, while remaining fully training-free and computationally efficient.

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

# A APPENDIX

**Language polishing statement.** Portions of this manuscript were refined for clarity, grammar, and style using a large language model (LLM). The scientific content, experimental design, results, and conclusions remain solely those of the authors.

## A.1 MATHEMATICAL FORMULATION OF COT-REFLECTION AND SDMRA

**Notation.** We provide a theoretical grounding for the proposed token-level local resampling in the Show-o framework.(Xie (2025)).We denote the generative distribution by $p_\theta(\mathbf{x} \mid \mathbf{y})$, where $\mathbf{x} = (x_1, \ldots, x_T)$ is the sequence of visual tokens of length $T$, and $\mathbf{y}$ is the textual condition. Each token index is $i \in \{1, \ldots, T\}$. The generation proceeds in discrete steps $t = 1, 2, \ldots$, following a MaskGIT(Chang et al. (2022)) style refinement. At a particular step $t^*$ we perform backtracking resampling. We use subscript $t$ to denote the predictive distribution at step $t$.

At step $t$, the model produces a probability distribution $p_t(i, \cdot)$ for each position $i$. The most probable token is

$$v_{t,i}^\star = \arg\max_{v \in \mathcal{V}} p_t(i, v), \tag{2}$$

with corresponding probability

$$\pi_{t,i} = \max_v p_t(i, v) = p_t(i, v_{t,i}^\star). \tag{3}$$

Let $\mathcal{K}_t$ denote the index set of tokens that are *preserved* (i.e., previously fixed and not to be re-sampled) at step $t$, and let $\overline{\mathcal{K}}_t = \{1, \ldots, T\} \setminus \mathcal{K}_t$ denote the masked positions to be predicted. The full intermediate sample used for semantic analysis is constructed by combining preserved tokens and argmax predictions:

$$\mathbf{x}_t^{\text{full}} = \left(x_{t,1}^{\text{full}}, \ldots, x_{t,T}^{\text{full}}\right), \quad x_{t,i}^{\text{full}} = \begin{cases} x_i^{\text{fixed}}, & i \in \mathcal{K}_t, \\ v_{t,i}^\star, & i \in \overline{\mathcal{K}}_t. \end{cases} \tag{4}$$

This composed image $\mathbf{x}_t^{\text{full}}$ is fed into the unified multimodal model (UMM) for semantic interpretation. The CoT-Reflection module returns (i) an refined prompt $\mathbf{y}_{\text{opt}}$ and (ii) an error-concept index set $\mathcal{O}_{\text{err}}$.

**Step 1: Multi-layer Attention Masking.** Let $\mathcal{L}$ be a set of selected Transformer layers (e.g., $\mathcal{L} = \{10, 11, 14, 16, 17, 18\}$). For each layer $l \in \mathcal{L}$ and position $i$, obtain an attention score $A_i^{(l)} \geq 0$, normalized as

$$\tilde{A}_i^{(l)} = \frac{A_i^{(l)}}{\sum_{j=1}^T A_j^{(l)}}. \tag{5}$$

Given threshold $\tau_l$, define the layer-wise mask

$$M^{(l)} = \{ i \mid \tilde{A}_i^{(l)} \geq \tau_l \}. \tag{6}$$

The attention-indicated candidate error set is

$$M_{\text{attn}} = \bigcup_{l \in \mathcal{L}} M^{(l)}. \tag{7}$$

Map CoT-Reflection's error concepts $\mathcal{O}_{\text{err}}$ to token indices $M_{\text{err\_obj}}$. The final error mask is obtained as a union:

$$\mathcal{M} = M_{\text{attn}} \cup M_{\text{err\_obj}}, \tag{8}$$

corresponding to a logical OR operation ($1 \vee 0 = 1$, $1 \vee 1 = 1$, $0 \vee 0 = 0$). This ensures that any position flagged by either signal is treated as erroneous.

**Step 2: Confidence Reweighting.** Assign reweighting constants:

$$\varepsilon_{\text{keep}} = 1 \times 10^{-8}, \tag{9}$$

$$\varepsilon_{\text{err}} = 1 \times 10^{-9}. \tag{10}$$

Define per-position factors:

$$\varepsilon_i = \begin{cases} \varepsilon_{\text{keep}}, & i \in \mathcal{K} \text{ (preserved)}, \\ \varepsilon_{\text{err}}, & i \in \mathcal{M} \text{ (error region)}, \\ 1, & \text{otherwise}, \end{cases} \tag{11}$$

where $\mathcal{K}$ denotes the global set of preserved tokens with $\mathcal{K} \cap \mathcal{M} = \varnothing$.

**Step 3: Combined Scoring and Sorting.** Define the adjusted score:

$$s_{t,i} = \pi_{t,i} \cdot \varepsilon_i. \tag{12}$$

Tokens are ranked by $s_{t,i}$ in descending order.

**Step 4: Noise Scheduling and Token Retention.** Given noise schedule $\{\beta_1, \ldots, \beta_T\}$, define

$$\bar{\alpha}_t = \prod_{s=1}^{t}(1 - \beta_s). \tag{13}$$

Let the retention ratio be

$$\rho(t) = f(\bar{\alpha}_t), \qquad f(x) = x^{\gamma}, \ \gamma \in (0, 1], \tag{14}$$

and the retained token count

$$k(t) = \lceil \rho(t)\, T \rceil. \tag{15}$$

**Step 5: Backtracking and Parallel Decoding.** At backtracking step $t_{\text{back}}$, compute $k = k(t_{\text{back}})$ and select the top-$k$ indices by $s_{t,i}$, denoted $\mathcal{K}_k$. Mask the complement $\overline{\mathcal{K}}_k$, initialize with mask symbols, and resume parallel decoding from $t_{\text{back}}$ conditioned on $\mathbf{y}_{\text{opt}}$, updating only positions in $\overline{\mathcal{K}}_k$.

This construction ensures that the intermediate image for CoT-Reflection preserves fixed tokens and fills masked regions with argmax tokens, enabling semantic analysis and targeted resampling in a MaskGIT-style pipeline.

**Computational Efficiency of Backtracking.** Directly resampling the entire image at a late generation step would require recomputing all $T$ token predictions in the MaskGIT pipeline, resulting in a complexity of $O(T)$ per step. In contrast, our local backtracking strategy focuses only on a subset of tokens $\overline{\mathcal{K}}_k$ identified as erroneous or uncertain. Let $|\overline{\mathcal{K}}_k| \ll T$ denote the number of masked positions to be re-sampled. Then, the computational cost per step reduces to $O(|\overline{\mathcal{K}}_k|)$, which is typically orders of magnitude smaller than $T$.

Formally, let $F_\theta(\mathbf{x}, \mathbf{y})$ denote the forward pass of the generative model. For global re-sampling:

$$\text{Cost}_{\text{global}} = O(T \cdot F_\theta),$$

while for local backtracking:

$$\text{Cost}_{\text{local}} = O(|\overline{\mathcal{K}}_k| \cdot F_\theta) \ll \text{Cost}_{\text{global}}.$$

Hence, local backtracking enables targeted refinement guided by semantic error signals from CoT-Reflection while avoiding redundant computation on tokens that are already correct. This approach balances efficiency and quality, allowing iterative improvement without the high cost of full image re-generation.

