# OpenReview forum: "CoT-RC: Chain-of-Thought Reflection and Correction for Image Generation without Extra Training"
_ICLR.cc/2026/Conference — ICLR 2026 Conference Withdrawn Submission_

### Official Review · Reviewer_bySL · 2025-10-28

**Soundness:** 2
**Presentation:** 2
**Contribution:** 2
**Rating:** 2
**Confidence:** 3

**Summary:**

This paper introduces a training-free CoT-enhanced framework for unified MLLMs. They propose CoT-Reflection module, using MLLM-generated interim visual descriptions and an LLM’s structured reasoning to detect semantic deviations and produce a refined prompt. Moreover, they propose a semantic-driven region alignment module fuses multi-layer cross-modal attention with CoT-identified objects to mask, backtrack, and resample only erroneous tokens, boosting control and efficiency without retraining.

**Strengths:**

1. **Training-free.** This approach is training-free and plug-and-play.
2. **Efficient and effective design.** The semantic-driven masked region alignment is efficient and region-focused. If the attention mask is correct, it can avoid modifying the correctly generated regions.

**Weaknesses:**

1. **Highly relies on attention masks.** The semantic-driven masked region alignment highly relies on the correctness of attention masks. If the musk is incorrect, this approach might harm the generation result. The author should discuss this problem or verify the correctness of the attention masks.

2. **Generalization.** The method involves many hyperparameters, which raises concerns that it may require careful tuning and may not generalize well to different image distributions.

3. **Limited baselines.** This paper conducts experiments only on Show-o, which does not validate the method’s generalization. I suggest adding experiments on other baselines, such as Janus Pro or Bagel.

**Questions:**

See weakness

---

### Official Review · Reviewer_P84W · 2025-10-30

**Soundness:** 2
**Presentation:** 2
**Contribution:** 2
**Rating:** 2
**Confidence:** 4

**Summary:**

This paper proposes a training-free framework, CoT-RC, that integrates chain-of-thought reasoning into unified multimodal models for image generation. The method introduces a global reflection module and a local semantic-driven correction module to iteratively refine generated images without additional training. Experiments on GenEval and T2I-CompBench claim significant improvements over the Show-o baseline and some existing CoT-based methods.

**Strengths:**

1. Interesting idea of leveraging CoT reasoning for image refinement in a training-free manner, which could potentially make multimodal reasoning more efficient.
2. The modular design (global reflection + local correction) is intuitively reasonable and well-structured.
3. The paper includes extensive experiments across multiple benchmarks.

**Weaknesses:**

1. Limited novelty. The method is a straightforward combination of prompt refinement and attention-based masking, conceptually similar to prior reflection or reinforcement-based methods such as T2I-R1[1] and ImageGen-CoT[2].
2. Lack of comparison with chain-of-thought image generation baselines such as [1][2].
3. The experiments mainly show modest quantitative gains and lack qualitative reasoning evidence.




[1] T2I-R1: Reinforcing Image Generation with Collaborative Semantic-level and Token-level CoT. Dongzhi Jiang, Ziyu Guo, Renrui Zhang, Zhuofan Zong, Hao Li, Le Zhuo, Shilin Yan, Pheng-Ann Heng, Hongsheng Li
[2]. ImageGen-CoT: Enhancing Text-to-Image In-context Learning with Chain-of-Thought Reasoning. Jiaqi Liao, Zhengyuan Yang, Linjie Li, Dianqi Li, Kevin Lin, Yu Cheng, Lijuan Wang

**Questions:**

same as weaknesses

---

### Official Review · Reviewer_ovHe · 2025-11-01

**Soundness:** 3
**Presentation:** 2
**Contribution:** 2
**Rating:** 4
**Confidence:** 3

**Summary:**

This paper proposes CoT-RC, a training-free Chain-of-Thought Reflection and Correction framework for image generation. The key idea is to plug a reasoning-reflection loop into a unified multimodal model (UMM) without retraining. The system uses two main modules:
(1) a CoT-Reflection module that performs step-by-step reasoning to detect semantic errors and refine prompts;
(2) a Semantic-Driven Masked Region Alignment (SDMRA) module that pinpoints and fixes local token-level errors.

**Strengths:**

Nice idea: The “training-free reasoning loop” is clever and quite fresh. It’s a practical way to integrate CoT reasoning into image generation without touching model weights.

Good structure: The paper is well organized, and the pipeline (reflection + correction + backtracking) makes sense. The figures help a lot.

Training-free and efficient: The idea of improving reasoning without retraining is appealing for large-scale models

**Weaknesses:**

Writing quality: The paper reads a bit stiff and translated in places. Some sentences are long and hard to follow. Polishing the language would make it much clearer.

Limited baselines: The evaluation focuses mostly on Show-o and a few AR models. It’d be more convincing if tested on a diffusion model or at least discussed.

No runtime or cost analysis: The method is described as “efficient,” but there’s no real runtime comparison or latency measurement.

Missing user perception metrics: Benchmarks like GenEval are great for compositional reasoning, but I’d also like to see FID, CLIPScore, or human ratings to check image quality.

Theory side a bit weak: The paper doesn’t clearly explain why CoT reasoning helps generation beyond intuition.

**Questions:**

How much does the performance depend on the quality of the reasoning model (e.g., Phi-4)?

What’s the typical number of reflection-correction iterations before convergence?

Is the CoT-Reflection ever misleading — e.g., overcorrecting or drifting away from the prompt?

How long does inference take compared to Show-o or PARM?

Could this method be plugged into diffusion models too?

---

### Official Review · Reviewer_efai · 2025-11-01

**Soundness:** 2
**Presentation:** 2
**Contribution:** 3
**Rating:** 4
**Confidence:** 4

**Summary:**

This paper proposes CoT-RC, a training-free framework that integrates Chain-of-Thought (CoT) reasoning into autoregressive image generation to improve accuracy and controllability. The approach leverages Unified Multimodal Models (UMMs) to perform both image-level global correction through semantic reasoning and token-level local correction through positional awareness. The framework introduces two core modules: (1) a CoT-Reflection module that analyzes intermediate visual tokens to identify semantic deviations and generate refined prompts, and (2) a Semantic-Driven Masked Region Alignment (SDMRA) module that localizes erroneous regions using attention maps for targeted correction. Experiments on GenEval and T2I-CompBench demonstrate improvements over the Show-o baseline (68% to 78% on GenEval) and competitive performance against larger models and reinforcement learning-based approaches.

**Strengths:**

1. The method requires no additional training, fine-tuning, or reward model design, making it computationally efficient and easily deployable compared to methods like PARM and ImageGen-CoT.
2. The combination of semantic-level reasoning (CoT-Reflection) with token-level local correction (SDMRA) is well-motivated and addresses limitations of prior CoT-based image generation methods that operate primarily at the text level.
3. The method achieves significant improvements across multiple benchmarks, with particularly notable gains in challenging tasks like counting (+19%) and position (+26%) on GenEval, demonstrating effectiveness in compositional reasoning.

**Weaknesses:**

1. The paper lacks analysis about the inference efficiency analysis of the method. For example, we don't see the average number of iterations of the model for a benchmark, and the convergence analysis. The performance vs iteration curve is also missing for a deeper understanding of the effectiveness of the method.
2. Critical parameters lack justification or sensitivity analysis. For example, why layers {10, 11, 14, 16, 17, 18} specifically? How was the maximum iteration limit of 10 chosen? etc.

**Questions:**

1. What is the actual computational overhead? Please provide wall-clock time comparisons and memory usage statistics compared to baseline Show-o and Show-o+PARM.
2. Can you provide justification about the hyper-parameter selection, including the transofmer layers, max number of iteration, attention thresholds.
3. Please provide efficiency analysis, as specified in weekness 1.
4. Line 167, typo "presents the propose"

---

### Note · Authors · 2025-12-11

I have read and agree with the venue's withdrawal policy on behalf of myself and my co-authors.